From trace to trace maker: Oligocene–Miocene coprolites of southern Poland and their potential producers

Brachaniec Tomasz tomasz.brachaniec@us.edu.pl 1
Środek Dorota 1
Salamon Mateusz 1
Bugajski Michał 2
Duda Piotr 3
Danielak Adam 4
Janiszewska Magdalena 4
Sadlok Grzegorz 1
Kuśnierczyk Wojciech 2
1 Faculty of Natural Sciences, University of Silesia in Katowice , Sosnowiec , Poland
2 Fossil Amateur Club “Inkluzja” , Łódź , Poland
3 Faculty of Science and Technology, University of Silesia in Katowice , Sosnowiec , Poland
4 Municipal Zoological Garden in Łódź , Łódź , Poland
De Baets Kenneth
Electronic publication date: 2025 Nov 3
Publication date: 2025
Volume: 13
Electronic Location ID: e20242
Received 2025 May 14; Accepted 2025 Sep 24
Copyright: ©2025 Brachaniec et al.
Copyright year: 2025
Copyright holder: Brachaniec et al.
License: This is an open access article distributed under the terms of the Creative Commons Attribution License, which permits unrestricted use, distribution, reproduction and adaptation in any medium and for any purpose provided that it is properly attributed. For attribution, the original author(s), title, publication source (PeerJ) and either DOI or URL of the article must be cited.
License URL: https://creativecommons.org/licenses/by/4.0/

Keywords: Terrestrial and marine bromalites, Coprolites, Faeces, Oligocene, Miocene, Poland

Funding: National Science Centre, Poland (ww.ncn.gov.pl) 2023/49/B/ST10/00322 This research was funded in whole by the National Science Centre, Poland (www.ncn.gov.pl), grant no. 2023/49/B/ST10/00322 for Tomasz Brachaniec. The funders had no role in study design, data collection and analysis, decision to publish, or preparation of the manuscript.

==============================
In this paper we describe coprolites from deep-marine Oligocene sediments, shallow- and deep-marine Miocene deposits, as well as Miocene continental environments in southern and central Poland. The Oligocene and Miocene coprolites from marine environments (hereinafter referred to as M) are classified into six morphotypes: (M1) sinusoidal, (M2) elongated and straight, (M3) curved, (M4) irregular, (M5) S-shaped, and (M6) oval. Sinusoidal coprolites, previously interpreted as originating from predatory fish (e.g., Palimphyes, Oligophus, and indeterminate taxa), are reinterpreted here, based on actualistic observations, as crustacean (crab) faeces. Morphotypes (M2)–(M5) are attributed to fish, while the oval type (M6) is tentatively linked to columbid-like birds, although alternative producers cannot be excluded. Miocene deep-sea coprolites are represented by relatively long, complex fecal masses composed of constricted strings, suggesting holothurians or cephalopods as potential producers. Elongated Miocene coprolites from shallow-water environments are likely to have been produced by teleost fish - most likely Sparidae - or by sharks. However, other vertebrates, including toothed and toothless cetaceans and porpoises, cannot be ruled out. The terrestrial Miocene specimens include five morphotypes (hereinafter referred to as T-terrestrial, T1–T7): (T1) oval, (T2) sinusoidal, (T3) elongated with two submorphotypes (T3/1–siderite individuals and T3/2–apatite individuals), (T4) S-shaped, and (T5) irregular. They are interpreted as coprolites likely produced by snakes (T1–T3/1, T4, T5), and small mammals such as Sciuridae and/or Chiropteridae (T3/2). Overall, these data provide new insights into the diversity of post-Mesozoic coprolites and refine our understanding of their producers and associated ecosystems in Central Europe.

Introduction

The oldest known vertebrate coprolites date back to the Ordovician (e.g., Hunt & Lucas, 2021). However, most published data on coprolites pertain to the Mesozoic era (e.g., Eriksson et al., 2011; Salamon et al., 2012; Schweigert & Dietl, 2012; Brachaniec et al., 2015; Schwimmer, Weems & Sanders, 2015; Zatoń et al., 2015; Niedźwiedzki et al., 2016; Vajda et al., 2016; Chin, Feldmann & Tashman, 2017; Segesdi et al., 2017; Barrios-de Pedro et al., 2018; Francischini, Dentzien-Dias & Schultz, 2018; Barrios-de Pedro, Chin & Buscalioni, 2020; Qvarnström et al., 2019; Qvarnström et al., 2024; Lukeneder et al., 2020; Rummy, Halaçlar & Chen, 2021; Román et al., 2024; Rozada et al., 2024).

Post-Mesozoic coprolites - or objects interpreted as such - have been attributed to a range of producers, including giant earthworms, fish, rodents, notoungulates, hathliacinid and borhyaenoid marsupials, hyenas and/or hyaenids and barbourofelids, as well as various indeterminate carnivorans, sirenians, and crocodilians. They have been reported from scattered localities across Europe, North and South America, and Asia (e.g., Wetmore, 1943; Amstutz, 1958; Edwards, 1976; Wilson, 1987; Richter & Baszio, 2001; Seilacher et al., 2001; Richter & Wedmann, 2005; Dvořák et al., 2010; Godfrey & Smith, 2010; Peñalver & Gaudant, 2010; Pesquero et al., 2011; Stringer & King, 2012; Hunt & Lucas, 2014; Dentzien-Dias et al., 2018; Wang et al., 2018; Collareta et al., 2019; Kapur et al., 2019; Tomassini et al., 2019; Abella et al., 2021; Gross et al., 2023; Román et al., 2024). A comprehensive overview of numerous Quaternary coprolites was provided by Wood & Wilmshurst (2014), Wood & Wilmshurst (2016), Tolar & Galik (2019), Agliano et al. (2024), Cambronero & García (2024), and Hunt & Lucas (2021); for review see also Gurjão et al. (2024) and literature cited therein.

The only marine coprolites from post-Mesozoic sediments of Poland were thoroughly described and illustrated by Bajdek & Bieńkowska-Wasiluk (2020), based on material from the Oligocene (Rupelian) of southeastern Poland. These authors documented sixteen coprolites from two localities within deep-water sediments of the Menilite Formation - an interval renowned for its spectacular fossil fish assemblages (e.g., Bieńkowska, 2004; Kotlarczyk et al., 2006; Bieńkowska-Wasiluk, 2010). Bajdek & Bieńkowska-Wasiluk (2020) concluded that the elongated, linear (drop-like), often strongly sinuous, and occasionally tear-shaped coprolites they described (see table 1 and fig. 2 in Bajdek & Bieńkowska-Wasiluk, 2020) were most likely produced by carnivorous teleost fish. Brachaniec et al. (2022) described 29 lacustrine, excrement-shaped ferruginous masses—referred to as “alleged” coprolites—from the Miocene (Burdigalian) deposits of the Turów lignite mine in southwestern Poland. According to these authors they were produced by turtles and snakes. The aim of this study is to describe numerous coprolites originating from both lacustrine and marine environments in Poland. The marine settings are represented by Oligocene and Miocene sediments from thirteen localities in southeastern Poland, while the studied lacustrine deposits are Miocene in age and come from southwestern, southern, southeastern, and central parts of Poland (Fig. 1). The coprolites have been categorized into distinct morphotypes. Their mineralogical composition, associated fossil inclusions, palaeoecological context, and the broader palaeobiological significance of the findings are discussed in detail.

Figure 1 Geological settings of studied locations.

(A) Map of Poland with marked research areas. (B) Kleszczów Graben area. (C) Southern edge of the Holy Cross Mountains. (D) Turów area. (E) Roztocze. (F) Menilite-Krosno Series of the Outer Carpathians. (G) Stratigraphic section and positions of sites where the coprolites have been documented. Compiled and slightly modified after: Kotlarczyk et al., 2006; Wysocka, Jasionowski & Peryt, 2007; Olchowy, Krajewski & Felisiak, 2019; Brachaniec et al., 2022; Salamon et al., 2024.

Geological setting

The field works were carried out in five areas located in southern and central Poland (see Fig. 1).

Kleszczów Graben area

The Kleszczów Graben is located in central Poland in Łódz Voivodeship; the graben is over 80 km long and up to three km wide (‘B’ on Fig. 1A). It is the deepest tectonic depression in the Polish Lowlands as it exceeds 550 m below sea level in depth (Widera, Klęsk & Urbański, 2024). Its bedrock is formed by Permian salts and carbonates of Jurassic to Cretaceous age (e.g., Olchowy, Krajewski & Felisiak, 2019). The tectonic development of the graben began in Cenozoic (Paleocene) and its in-filling sediments experienced three main phases of deformation, including Valachian stage, Bełchatów stage and “upper” stage with galcitectonics (Krzyszkowski, 1989) and Rupelian (early Oligocene). The palaeotectonic evolution of this graben accelerated following the late Oligocene (Chattian) regional uplift. The lowermost Miocene sediments are siliciclastics consisting of sands, muds, clays, and thin layers of lignite (Czarnecki, Frankowski & Kuszneruk, 1992). A coal complex of lignite follows these lowermost siliciclastics of Miocene and comprises lenses of non-coal sediments and rocks, including sands, clays, lacustrine chalk, flints, sandstones, and paratonsteins (tuff horizons; Widera, Klęsk & Urbański, 2024). The middle Miocene succession ends with clay-coal and clayey sands complexes as seen in the Bełchatów section - these complexes form a total thickness of up to 100–150 m (Widera, Klęsk & Urbański, 2024) and provide fossil plant remains and the coprolites described herein.

Southern Poland (southern edge of the Holy Cross Mountains)

Miocene sediments exposed in the southern edge of the Holy Cross Mountains are located in the marginal, northern part of the Carpathian Foredeep (‘C’ on Fig. 1A). This area was located in the northern part of the central Paratethys in the Miocene (Salamon et al., 2024). The coastal and shallow-marine sediments of the area have formed in an environment of moderate environmental energies (Studencki, 1999). Occasionally, the sediments were influenced by storms, which resulted in bivalve accumulations with numerous other fossils (Bałuk & Radwański, 1977; Gutowski, 1984). Abundant, large foraminifers (Amphistegina and Heterostegina) are typical for these shallow marine early Badenian Paratethys deposits. No structures indicative of linear currents have been observed, which might be an indication of high turbulence waters during the storms. One coprolite specimen comes from the so-called Heterostegina Sands of the Pińczów Formation of Gołuchów locality with common foraminifers, molluscs, bryozoans, serpulids, echinoderms, and teeth of fish (Salamon et al., 2024).

Lithified lower Kimmeridgian oolitic-bioclastic limestones are exposed at the Gołuchów site and fine-grained red-algal sandy limestones with isolated pebbles of the same Kimmeridgian oolitic limestones cover them.

South-western Poland (Turów area)

The Turów lignite mine (‘D’ on Fig. 1A) is located in the south-western part of the Lower Silesia Voivodeship (south-western Poland). It covers the former village of Turów (near Bogatynia), in the central part of the mesoregion Żytawa-Zgorzelec Depression located between the state borders of Poland, Czechia and Germany. The thickness of the sediments exposed in the Turów profile is about 250 m. These sediments comprise seven lithostratigraphic units of sedimentary rocks. Most of those units are dominated by clays and/or muds with only minor intercalations of coarser facies, like sands or gravel-bearing sands (Kasiński et al., 2015). The oldest Cenozoic sediments of the sedimentary succession exposed herein are Oligocene sediments (Egger age), forming the lower and middle part of the Turoszów Formation (Kasiński et al., 2015). There are coal seams in the middle part of the profile. These seams belong to the Opolno and the Biedrzychowice Formations, which are the primary deposits exploited by the Turów mine. The coprolites described in the current study have been collected from the upper part of the Biedrzychowice Formation (Karpatian, Burdigalian; comp. (Brachaniec et al., 2022). The youngest sediments are of the Gozdnica Formation and Pleistocene till of glacial origin. These units are, contrary to the older ones, dominated by sands and gravels (Kasiński et al., 2015).

South-eastern Poland (Roztocze)

The Roztocze is a geographical region in south-eastern Poland located in the Lubelskie and partly in the Podkarpackie Voivodeships. It connects the Lublin Upland to the Podolia in Ukraine (‘E’ on Fig. 1A). Miocene sediments of the Roztocze are dated as Badenian and Sarmatian (Wysocka, Jasionowski & Peryt, 2007). Although these are marine formations, determining their exact age is challenging due to the peculiarities of the depositional environment and the complex connections between the Pre-Carpathian foredeep basin and the Central and Eastern Paratethys. The use of separate lithostratigraphic schemes by Polish and Ukrainian geologists for cross-border strata further complicates age determinations (Bogucki et al., 1998). The investigated Miocene sediments represent diversified shallow-marine and shoreface facies: quartz sands dominate and are overlain by pelitic limestones in the lower part, and quartz-rodoid sands, organodetritic limestones, reef-type organodetritic limestones, shells, marls and serpulid-microbialitic limestones (Musiał, 1987; Jasionowski, 1997). Current field investigations focused on four sites (Brusno, Huta Różaniecka, Józefów, and Żelebsko; for details see e.g., Wysocka, Jasionowski & Peryt, 2007). Coprolites were found in Sarmatian calcarenites with spheroidal bodies of serpulid-microbial limestones at the Żelebsko site (Fig. 1E).

South-eastern Poland (Menilite-Krosno Series of the Outer Carpathians)

The Menilite-Krosno Series of the Outer Carpathians is located in southeastern Poland in the Subcarpathian Voivodeship (‘E’ on Fig. 1F). At the Eocene–Oligocene boundary, tectonic activity and eustatic drop of sea level resulted in restriction of contact between sedimentary sub-basin of the Menilite-Krosno Series of the Outer Carpathians (part of the central Paratethys) and larger basin of the eastern Paratethys and of the Mediterranean domain (Popov et al., 2002). The Menilite-Krosno Series of Oligocene (Rupelian and Chattian) and Miocene (Aquitanian and Burdigalian) comprise bituminous marlstones, cherts, shales, and sandstones with common fish fossils (e.g., Bieńkowska-Wasiluk, 2010). The series is a result of the activity of submarine fans, bottom currents, and deposition from low concentration turbidity currents as well as pelagic sedimentation and blooms of coccolithophores (Kotlarczyk et al., 2006). Current fieldworks focused on 24 sites of several hundred listed by Kotlarczyk et al. (2006) (Table S1, Fig. S1), which represent both Oligocene and Miocene sediments. The studied coprolites were found in nine of the selected sites (Oligocene: Kąkolówka I, Kąkolówka II, Wola Czudecka, Futoma, Jamna Dolna, Rudawka Rymanowska, Równe, Wujskie, and Jasienica Rosielna; Miocene: Temeszów and Brzuska; for detailed geology and lithology of these localities see (Kotlarczyk et al., 2006).

Materials and Methods

Studied material coprolites are housed in Sosnowiec at the Institute of Earth Sciences, Faculty of Natural Sciences of the University of Silesia in Katowice, Poland (hereafter: IES), and catalogued under the registration numbers GIUS 10–3796/O/1–300 (for Oligocene) and GIUS 10–3796/M/1–34 (for Miocene). Detailed specimen lists and descriptions are provided in Tables S1 and S2.

Fossil fishes and specimens of potential producers illustrated in Figs. S3–S5 are from the Instiute of Earth Sciences, Sosnowiec, Poland (catalog acronyms IES) and the Museum of Fossils and Minerals, Dubiecko, Poland and have catalog numbers starting with acronyms Kr., MSMD, ROJ, RORR, Ma, ROL, ROJR, ROU, ROM (all are collected in the Museum of Fossils and Minerals, Dubiecko, Poland).

There have been 18 coprolites studied from the Kleszczów Graben area (continental Miocene; GIUS 10–3796/M/1–5, 6, 6(1), 6(2), 6(3), 6(4), 6(5), 7–12) and five of those specimens have been selected for detailed investigation in thin sections (GIUS 10–3796/M/1, 2, 6, 7, 11). The Turów area (continental Miocene) provided 18 more specimens (GIUS 10–3796/M/14–31), and 3 of those have been subjected to further examination in thin sections (GIUS 10–3796/M/17, 20, 27). The single specimen (GIUS 10–3796/M/13) collected from the southern edge of the Holy Cross Mountains (marine Miocene), and another one from the Roztocze area (GIUS 10–3796/M/32), have been also selected for thin section analyses. 302 specimens of the Menilite-Krosno Series were studied (marine Oligocene and Miocene; GIUS 10–3796/O/1–300, GIUS 10–3796/M/33,34), among which 50 were designated to be further studied in thin-sections (GIUS 10–3796/O/1–47, GIUS 10–3796/O/107, GIUS 10–3796/O/294, GIUS 10–3796/O/300, GIUS 10–3796/M/33, 34). Nearly all specimens were macroscopically documented in situ through field photography during field investigations. An exception was the group of elongated specimens with a distinct, prominently pointed end [GIUS 10–3796/M/6, 6(1), 6(2), 6(3), 6(4), 6(5)], which were recovered over clay washing. 2 bulk samples were collected in the field, weighing 40 kg and 45 kg, respectively. These samples were transported to the laboratory in Sosnowiec (Poland) belonging to the IES. The samples were washed using running hot tap water, screened on a column (Ø3.0, 1.0, 0.315 and 0.1 mm-mesh respectively), and finally dried at 150 °C. This washed, screened and dried residue was observed under a Leica WildM10 microscope in search for microremains.

The coprolites described in this article have been futher investigated with a number of different analytical tools. The methodological details are presented below.

Optical microscopy and thin-sectioning

Optical observations of thin sections have been carried out using Leica SZ-630T dissecting microscope and Nikon Eclipse E100 light microscopy, while the microphotographs have been collected using Olympus BX51—a polarizing microscope equipped with an Olympus SC30 camera and a halogen light source (IES).

Thin sections were performed in the Grindery at the IES. Specimens were embedded in Araldite epoxy resin, sectioned, mounted on microscope slides and polished using silicon carbide andaluminum oxide powders until reaching 30 µm in thickness.

Scanning electron microscopy

The chemical composition of the coprolite matrix and embedded microfossils have been examined using the desktop scanning electron microscope (SEM) Phenom XL (Thermo Fisher Scientific, Netherlands), equipped with integrated energy-dispersive X-ray spectroscopy (EDS) detector and secondary electron detector (SED), IES. Observations were conducted under low-vacuum conditions at 15 kV voltage, without coating.

For this study, one representative of each morphotype was selected. These were specimens with the following acronyms: GIUS 10–3796/O/1, GIUS 10–3796/O/3, GIUS 10–3796/O/6, GIUS 10–3796/O/11, GIUS 10–3796/O/13, GIUS 10–3796/O/20, GIUS 10–3796/M/3, GIUS 10–3796/M/5, GIUS 10–3796/M/6(1), GIUS 10–3796/M/9, GIUS 10–3796/M/12, GIUS 10–3796/M/18 (for details see Tables S1 and S2).

Microtomography

One representative specimen from six identified morphotypes was selected for virtual sectioning (specimens GIUS 10–3796/O/2, GIUS 10–3796/O/9, GIUS 10–3796/O/18, GIUS 10–3796/O/21, GIUS 10–3796/O/30, GIUS 10–3796/O/111, GIUS 10–3796/M/3, GIUS 10–3796/M/6, GIUS 10–3796/M/9, GIUS 10–3796/M/12, GIUS 10–3796/M/13, GIUS 10–3796/M/18, GIUS 10–3796/M/21, GIUS 10–3796/M/32, GIUS 10–3796/M/34).

In microtomographic studies, the flat shape of the samples in the form of a disc makes it difficult to optimally position them in relation to the radiation source and the detector. Precise positioning is also required so that the X-ray beam penetrates the entire thickness of the sample without losing focus. Incorrect positioning leads to image distortions (artefacts) caused by differences in the thickness of the X-rayed layers and to difficulties in 3D reconstruction due to the limited number of projection angles. Due to these difficulties some of the samples had to be cut using a mini-grinder. The form of columns facilitates imaging using an X-ray scanner.

Microtomographic studies were carried out in the Laboratory of Computed Microtomography of the Institute of Biomedical Engineering of the University of Silesia in Katowice. The samples were scanned at voltage parameters of 160 kV and current of 50 µA, 100 µA with resolutions of eight µm, 10 µm and 25 µm. Each projection with a resolution of 2,024 × 2,024 pixels consisted of three repetitions with an exposure time of 500 ms. The scanning time of the coprolites was about one hour during which 2100 X-rays were taken.

The images after reconstruction were processed using Volume Graphics®VGSTUDIO Max software, where image normalization and appropriate positioning and geometric measurements were performed. Visualization, animations and detailed analysis were performed using the Volume Graphics®myVGL viewer.

The raw data (image stacks) and software for viewing them are available here: https://zenodo.org/records/16742330 (DOI 10.5281/zenodo.16742330).

Observations of modern excrements

For comparative observations, more than 400 faeces of extant animals were collected. The excrements belonged to crustaceans (crabs) and a diversity of vertebrates (fish, reptiles, birds, and mammals). They were collected at the Municipal Zoological Garden in Łódź, Poland, by staff there. The excrements were not removed from the animals’ natural enclosures where they were photographed. For comparative purposes, we also used archived data on the faeces of fish, amphibians, reptiles, birds, and mammals, which were collected in 2021 at the Silesian Zoological Garden in Chorzów, Poland (for details see (Brachaniec et al., 2022). Some specimens were photographed on private farms located in southern Poland; several forms produced by wild animals were observed in local forests. Particular attention was given to those clades that have representatives in the Oligocene and Miocene sediments of Poland and neighbouring areas, and could therefore have been among the producers responsible for the studied coprolites.

The excrements of the following animals were collected (current data and those from 2021 published in Brachaniec et al., 2022); (1) invertebrates: (a) hermit crab (Coenobita brevimanus), (b) flying crab (Liocarcincus holsatus). Excrements of sea cucumber (Holothuria sp.) and cephalopod (Nautilus pompilius) are redrawn from Knaust & Hoffmann (2020). (2) vertebrates; (2.1) fish: (a) Syngnathidae, (b) great barracuda (Sphyraena barracuda), (c) Perciformes, (d) Centriscidae (Aeoliscus strigatus), (e) Lobotiformes (Datnioides microlepis), (f) leopard shark (Stegostoma fasciatum), (g) brownbanded bamboo shark (Chiloscyllium punctatum); (2.2) reptiles: (a) king python (Python regius), (b) tiger python (Python molurus), (c) reticulated python (Malayopython reticulatus), (d) common boa (Boa constrictor), (e) king cobra (Ophiophagus hannah), (f) Korean ratsnake (Elaphe anomala), (g) common European viper (Vipera berus), (h) komodo dragon (Varanus komodoensis), (i) Mediterranean tortoise (Testudo hermanni), (j) steppe tortoise (Testudo horsfieldii), (k) Indian star tortoise (Geochelone elegans), (l) Spanish pond turtle (Mauremys leprosa), (m) Nile soft shell turtle (Trionyx triunguis); (2.3) birds: (a) Seba’s short-tailed bat (Carollia perspicillata), (b) house sparrow (Passer domesticus), (c) city pigeon (Columba livia forma urbana), (d) white-tailed Eagle (Haliaeetus albicilla), (e) clawless (Rollulus rouloul); (2.4) mammals: (a) brown hare (Lepus europeaus), (b) European mole (Talpa europaea), (c) Guinea pig (Cavia porcellus), (d) Swinhoe’s striped squirrel (Tamiops swinhoei), (e) European beaver (Castor fiber), (f) African lion (Panthera leo), (g) cheetah (Acinonyx).

Results

Coprolite morphology

A total of 339 coprolites were collected: 300 from Oligocene and 39 from Miocene sediments (for details and summary see Tables S1–S4. The Oligocene and Miocene marine coprolites are classified into six morphotypes: (M1) sinusoidal, (M2) elongated and straight, (M3) curved, (M4) irregular, (M5) S-shaped, and (M6) oval (see Figs. 2, 3).

Figure 2 Examples of coprolites collected in the Oligocene and Miocene marine sediments of Poland.

Kąkolówka I: (A) GIUS 10–3796/O/2. Curved morphotype; (B) GIUS 10–3796/O/7. Curved morphotype; (C) GIUS 10–3796/O/23. Sinusoidal morphotype; Kąkolówka II: (D) GIUS 10–3796/O/154. Sinusoidal morphotype; (E) GIUS 10–3796/O/181. Sinusoidal morphotype; (F) Kąkolowka I, GIUS 10–3796/O/60. Curved morphotype; (G) Kąkolówka I, GIUS 10–3796/O/77. Curved morphotype; Wola Czudecka: (H) GIUS 10–3796/O/251. Elongated morphotype; (I) GIUS 10–3796/O/253. Elongated morphotype; (J) GIUS 10–3796/O/259. Elongated morphotype; (K) GIUS 10–3796/O/274. Elongated morphotype; (L) Futoma, GIUS 10–3796/O/279. Oval morphotype; (M) Futoma, GIUS 10–3796/O/282. Irregular morphotype; Kąkolówka I: (N) GIUS 10–3796/O/96. Irregular morphotype; (O) GIUS 10–3796/O/98. Irregular morphotype; (P) GIUS 10–3796/O/107. S-shaped morphotype; (R) GIUS 10–3796/O/111. S-shaped morphotype; (S) GIUS 10–3796/O/135. S-shaped morphotype; (T) Jamna Dolna, GIUS 10–3796/O/294. Elongated morphotype; (U) Kąkolówka I, GIUS 10–3796/O/139. Oval morphotype. Scale bars five mm.

Figure 3 Examples of coprolites collected in Oligocene and Miocene marine.

Marine (A–D) and non-marine (E–N) sediments of Poland. Równe: (A) GIUS 10–3796/O/297; Jasienica Rosielna. Oval morphotype; (B) GIUS 10–3796/O/299; Kąkolówka I. Sinusoidal morphotype; (C) GIUS 10–3796/O/144; Temeszów. Irregular morphotype; (D) GIUS 10–3796/M/33; Gochułów. Elongated morphotype; (E) GIUS 10–3796/M/13; Roztocze area-Żelebsko. Elongated morphotype; (F) GIUS 10–3796/M/32; Turów area. Elongated morphotype; (G) GIUS 10–3796/M/16. Elongated morphotype; (H) GIUS 10–3796/M/19. Curved morphotype; (I) GIUS 10–3796/M/23. Irregular morphotype; (J) GIUS 10–3796/M/28. Curved morphotype; (K) GIUS 10–3796/M/30. Curved morphotype; Bełchatów (L) GIUS 10–3796/M/2. Irregular morphotype; (M) GIUS 10–3796/M/6. Elongated morphotype; (N) GIUS 10–3796/M/11. Oval morphotype. Scale bars five mm.

(M1) This type of coprolites is represented by strongly elongated forms with a maximum length not exceeding 56 mm and a diameter not exceeding 15 mm; they are more or less bent, compressed, clearly sinusoidal. Within them, macroscopic remains of vertebrates, most likely fish, are common, which are represented by bones and scales;

(M2) The coprolites here are elongated with a maximum length of 46 mm and a diameter not exceeding 17 mm; compressed; in contrast to morphotype M1, they are not sinusoidal and/or curved. Remains of other vertebrates, most likely fish represented by bones and scales are much rarer here than in morphotype M1.

(M3) They are similar to morphotype M2 with the difference that they are strongly bent, usually having the shape of the letter C; they are also much longer, because the longest individuals reach almost 60 mm in length and their diameter does not exceed several mm; compressed. Remains of other vertebrates, most likely fish represented by bones and scales are much rarer here than in morphotype M1 and comparable in quantity to those observed in morphotype M2.

(M4) This morphotype is represented by highly diversified specimens, of various sizes and shapes: from more or less regular ones inscribed in rectangles, triangles, less often squares, through irregular forms to slightly elongated ones. Most of them are compressed. This morphotype was divided into two subtypes: a) fragmentarily preserved coprolites (comp. Figs. 2M, 2N), which originally most likely belonged to the one of elongated morphotypes. Their characteristic feature is that they consist exclusively of remains of other vertebrates, most likely represented by bones and scales; b coprolites of various sizes and shapes, slightly elongated, apatite, and containing rare remains of fish bones and scales visible only microscopically (see Fig. 2O).

(M5) The coprolites belonging to this morphotype are elongated and have the shape of the letter S. Their length does not exceed 60 mm, and their diameters can reach up to 40% of the length. They are not compressed and remains of other vertebrates are relatively common here. (M6) They are more or less oval in shape, convex, and their diameters do not exceed 32 mm. Within them, the remains of other vertebrates are not visible macroscopically.

The terrestrial Miocene specimens include five morphotypes: (T1) oval, (T2) sinusoidal, (T3) elongated with two submorphotypes (T3/1 –siderite individuals and T3/2 –apatite individuals), (T4) S-shaped, and (T5) irregular.

(T1) This type is represented by a siderite mass having a more or less oval shape with diameters not exceeding 40 mm. They have a rough surface on which rare coalified debris is visible;

(T2) Coprolites classified in this type are siderite masses that are distinctly elongated and sinusoidal. Their maximum length is 54 mm, and the diameter does not exceed 23 mm. Similarly to the T1 morphotype, their surface is rough, and occasionally coalified debris is observed on it;

(T3) They are similar to morphotype T2, but they are not sinusoidal. Their maximum length is 65 mm, and the diameter is several times smaller. Siderite individuals are classified as submorphotype T3/1, while apatite individuals are classified as T3/2. The surface of submorphotype T3/1 is rough and may be covered by coalified debris, while the surfaces in T3/2 are smooth and no fauna has been found inside or on their surfaces.

(T4) Siderite individuals belonging to this morphotype are elongated and have a distinct S-shape. Their maximum length is 48 mm, and their diameter does not exceed 20 mm. Their surface is rough; coalified debris occasionally occurs on its surface.

(T5) Siderite individuals represented by highly diversified specimens, of various sizes and shapes: from more or less regular ones inscribed in rectangles or squares. Their diameter not exceed 47 mm. Their surface is rough; coalified debris occasionally occurs on its surface.

The colours of coprolites varied, even within the same morphotype and age group. Oligocene (M-KS) sinusoidal forms were most often black (51%) and brown (49%). Black (43%), brown (37%), grey (19%), and red (2%) specimens were found also among elongated Oligocene coprolites. The oval and the regular ones were grey (77%), red (21%), and pastel (2%) in colour. The S-shaped coprolites were black (60%), brown (30%), and red (105). Finally, the curved forms were red (70%), brown (25%), and grey (5%). The sediment in which the coprolites are found is light, ranging from light pastel, through light pink, light brown, light yellow to light graphite. The sediments hosting the purported coprolites are characterized by their light colors, ranging from whitish and creamy to light orange. The detrital particles are typically subangular to subrounded, with a dominant grain size below very fine silt (determined visually and by SEM). These sediments are here classified as calcilutites (Folk, Andrews & Lewis, 1970)—which correspond to mudstones in the classifications of Dunham (1962), Embry III & Klovan (1971), and Wright (1992). The sediments are finely laminated (millimeter-scale laminae), showing no visible biogenic vertical sediment mixing. The grain size, color, and degree of mineralization vary slightly between laminae. This variation in cementation may be attributed to differences in grain size, pore space, and/or mineralogical composition. However, the detailed characterization of the cementing phase was not the primary aim of this study. The parting surfaces between laminae bear coprolites and show no trace fossils. The laminae boundaries are typically sharp, though local, weak erosion of laminae is apparent in some areas.

In the case of continental Miocene specimens (Turów area), their colours varied from pale orange, through greenish red, to burgundy-colored. The ferruginous specimens from Kleszczów Graben were celadon, brown-blue, and locally red. Six specimens were light pastel to light brown. Specimens from the marine Miocene of Roztocze area and Gołuchów quarry (the edge of the Holy Cross Mountains) were light orange and light brown, respectively.

Microtomographic, optical and SEM microscopy studies

In all specimens from Menilite-Krosno Series of the Outer Carpathians, some undigested food remains were observed, and these food item remnants include mostly remains of fish (see e.g., Movie S1).

Thin sections of continental Miocene coprolites were analyzed in transmitted and reflected light. A dark, nearly opaque matrix can be seen in the specimens from Kleszczów Graben area (GIUS 10–3796/M/1, GIUS 10–3796/M/2, GIUS 10–3796/M/7, GIUS 10–3796/M/11) and from Turów area (GIUS 10–3796/M/17, GIUS 10–3796/M/20, GIUS 10–3796/M/27). The mineral matrix is homogeneous and some elongated structures can be observed within it. These elonged features have arcuate shapes in some cases and they appear to be light-reduction areas in reflected light whereas the surrounding matrix was oxidized. The dark (rusty, brown to almost black), slightly transparent colour of the matrix suggests an iron-rich mineral(s) that formed the matrix. A bright matrix can be observed in one specimen when seen under transmitted light ((GIUS 10–3796/M/6) –Kleszczów Graben area). No biogenic remains were observed in this case, only some indeterminate mineral structures. Similar results of thin section analyses were obtained from the specimens collected from the southern edge of the Holy Cross Mountains (marine Miocene; (GIUS 10–3796/M/13) and Roztocze area (GIUS 10–3796/M/32).

A bright and opaque matrix can be observed in thin sections made from the marine Oligocene and Miocene coprolites of the Menilite-Krosno Series of the Outer Carpathians (GIUS 10–3796/O/1–47, GIUS 10–3796/O/107, GIUS 10–3796/O/294, GIUS 10–3796/O/300, GIUS 10–3796/M/33, 34). The matrix is homogeneous in most of the analyzed samples, however in same cases small structures with angular edges can be noted. Numerous fish remains can be observed embedded within the matrix, and these remains, after further examination under SEM (Figs. 4 and 5), have been found to represent fish bones, scales and teeth.

Figure 4 BSE images of investigated coprolites from Oligocene coprolites of the Menilite-Krosno Series of the Outer Carpathians.

(A–E, J) Fish bones. (F–I?) Scales. (K–L) Teeth. (A–B, D, E, G–I) Kąkolówka I locality, GIUS 10–3796/O/107; (C) Jasienica Rosielna locality, GIUS 10–3796/O/300; (F, J–L) Jamna Dolna locality, GIUS10–3796/O/294. Scale bars 30 um.

Figure 5 BSE images showing unidentified fossil bone remains embedded with in coprolite matrix from Miocene of the Menilite-Krosno Series of the Outer Carpathians (GIUS 10–3796/M/33 and 34 respectively).

(A) T he coprolite/matrix boundary and the surrounding sediment, with bone fragments visible in both. (B) Remains of different morphology. (C–E) Close-ups of selected fossilized fragments. Scale bars 200 um.

Mineralogical and structural analyses

The chemical composition of coprolite no. GIUS 10–3796/M/33 was characterized by SEM and energy-dispersive X-ray spectroscopy (EDS) analyzes, which revealed that the coprolite matrix is highly porous and composed predominantly of microcrystalline fluorapatite. The fluorapatite occurs in small (approximately 0.5–4 µm in diameter), thin-walled vesicular structures. These morphologies are interpreted as mineral pseudomorphs after organic components originally present in the faeces, possibly including bacterial cells (Hollocher, Hollocher & Keith Rigby Jr, 2010). Some studies have proposed that such structures may be related to spherical bacteria, such as Enterococcus faecalis, and other cocci commonly found in fecal matter (Hollocher, Hollocher & Keith Rigby Jr, 2010). Experimental and natural observations indicate that bacteria and their enzymatic activity (e.g., phosphatases) can promote the precipitation of microcrystalline apatite (Hirschler, Lucas & Hubert, 1990; Lucas & Prevot, 1991; Jehl & Rougerie, 1995), suggesting that the original fecal microbiota may have played a role in the diagenetic mineralization process. Within the porous matrix, fragments of clearly organic origin composed of fluorapatite were identified (Figs. 4 and 5). These microfossils most likely represent bone fragments, teeth, and remnants of plant tissues. Additionally, the matrix contained non-organic mineral grains such as quartz and zircon, as well as crystals that formed within the coprolite voids, including calcite and framboidal pyrite. The only coprolite with a different chemical composition was one specimen from Turów. This specimen had also porous matrix structure but it consists of iron oxides and hydroxides. No microfossils were found within it.

Newly descibed modern faeces

The visual comparison made it possible to exclude modern faeces that differed significantly from the analyzed coprolites in terms of size and shape. These faeces samples were not taken into account in further analyses. The subsequent observations were based on a morphological comparison between the selected recent faeces and the studied coprolites. Surprisingly, crabs (Coenobita brevimanus) were observed to produce fecal masses of sinusoidal morphology (Fig. 6J) similar to coprolites to the herein described Oligocene coprolites (see e.g., Figs. 2C–2E, 3B). Nearly identical sinusoidal faeces (see Fig. 6K) were produced by another crab (flying crab, Liocarcincus holsatus), which is closely related to fossil representatives of Liocarcinus—a taxon commonly found in the Menilite-Krosno Series of the Outer Carpathiansno (Fig. S1). So far, this type of coprolite morphology has been attributed to predatory fishes (e.g., Bajdek & Bieńkowska-Wasiluk, 2020). However, despite the examination of numerous faces produced by extant fish taxa (a total of 30 species belonging to Scombriformes and Gadiformes), no corresponding sinusoidal morphology has been observed in the fecal remains of any of these taxa. The observed recent faeces of studied fish taxa were dominated by masses with morphologies resembling coprolites’ morphologies classified into straight, curved, and S-shaped categories (see Fig. 6N). These fish-produced fecal masses comprised various remains of other, presumably consumed fish individuals (bones, scales, teeth). Noteworthy, the studied coprolites with similar morphologies also contain fossil fish remnants.

Figure 6 Recent faeces.

(A) Brown hare (Lepus europeaus). (B) European mole (Talpa europaea). (C) Guinea pig (Cavia porcellus). (D) Swinhoe’s striped squirrel (Tamiops swinhoei). (E) Seba’s short-tailed bat (Carollia perspicillata). (F) House sparrow (Passer domesticus). (G) Syngnathidae. (H) City pigeon (Columba livia forma urbana). (I) Great barracuda ( Sphyraena barracuda). (J) Hermit crab ( Coenobita brevimanus). (K) Flying crab (Liocarcincus holsatus). (L) Seacucumber (Holothuria sp.; redrawn from Knaust & Hoffmann, 2020). (M) Cephalopod (Nautilus pompilius; redrawn from Knaust & Hoffmann, 2020). (N) Perciformes. Scale bars one cm.

Current observations show that barracudas produce more or less regular faeces, sometimes slightly tapering on one side (comp. Fig. 6I). There is a similar morphological type in the studied sample of Oligocene coprolites (more or less regular with macroscopically visible vertebrate remains; Figs. 2O, 3C). It is likely, based on morphologic and size criteria, that this fossil coprolite specimen was also produced by barracuda or related fishes (Sphyraena).

Oval and relatively large coprolites from the Oligocene marine sediments (Fig. 2U) do not contain any faunal remains. Their shape and size resemble the fecal masses produced by members of the bird family Columbidae (Fig. 6H; cf., Fig. S6). Noteworthy, fossil remains of these birds have been documented in the Menilite-Krosno Series of the Outer Carpathians (Bocheński, Tomek & Świdnicka, 2010).

Deep-sea coprolites documented from the Miocene deposits are represented by relatively long and complex faecal masses consisting of string with frequent constrictions (Fig. 3D). These fossil specimens have morphology most closely resembling feaces of holothurians (Holothuria sp.; Fig. 6L) and cephalopods (Nautilus pompilius; Fig. 6M).

The last type of bromalites compared with recent fecal masses consists of phosphatic specimens recovered from continental Miocene strata. These coprolites are elongated and exhibit a characteristic, prominently pointed end, likely formed as the anus contracted to close and sever the expelled fecal mass (Fig. 3M). Among vertebrates inhabiting the present-day terrestial environments of central Poland, the faeces of Sciuridae and Chiropteridae are most comparable to the fossil specimens, as they are similarly small and display a distinct pointed termination at one end (Figs. 6D, 6E); for summary see Table 1.

Table 1 List of animals whose faeces were examined during the current study.

Producer	Average dimensions in mm (length/width/diameter)	Shape	Schematic drawing	Source	
Sea cucumber; Holothuria sp.	10/2/-	Curved, elongated		Knaust & Hoffmann (2020)	
Cephalopod; Nautilus pompilius	15/7/-	Elongated, S-shaped		Knaust & Hoffmann (2020)	
Hermit crab; Coenobita brevimanus	14/4/2	Curved		Authors observation	
Flying crab; Liocarcincus holsatus	13/3/2	Sinusoidal		Authors observation	
Fish; Syngnathidae	3-30/3/1	Elongated, curved, oval		Authors observation	
Fish; Zebra moray; Gymnomuraena zebra	21/18/16	Oval		Authors observation	
Fish; great barracuda; Sphyraena barracuda	17/16/13	Oval		Authors observation	
Fish; Perciformes sp.	3-32/2/1	Elongated, curved, sinusoidal, oval		Authors observation	
Fish; Centriscidae; Aeoliscus strigatus	13/3/1	Irregular		Authors observation	
Fish; Lobotiformes; Datnioides microlepis	11/6/4	Elongated		Authors observation	
Fish; leopard shark; Stegostoma fasciatum	42/15/11	Elongated		Authors observation	
Fish; brownbanded bamboo shark; Chiloscyllium punctatum	26/12/10	Elongated		Authors observation	
Reptile; Mediterranean tortoise; Testudo hermanni	40-52/13/13	Elongated, curved		Brachaniec et al. (2022)	
Reptile; Indian star tortoise; Geochelone elegans	33/12/9	Curved, irregular		Brachaniec et al. (2022)	
Reptile; steppe tortoise; Testudo horsfieldii	24/9/4	Elongated, curved		Authors observation	
Reptile; Spanish pond turtle; Mauremys leprosa	21/8/5	Elongated		Authors observation	
Reptile; Nile soft shell turtle; Trionyx triunguis	16/7/5	Elongated		Authors observation	
Reptile; king pyton; Python regius	33/31/30	Oval, irregular		Brachaniec et al. (2022)	
Reptile; common boa; Boa constrictor	30/28/27	Oval		Brachaniec et al. (2022)	
Reptile; tiger python; Python molurus	38/12/8	Elongated		Authors observation	
Reptile; reticulated python; Malayopython reticulatus	32/8/6	Elongated		Authors observation	
Reptile; komodo dragon; Varanus komodoensis	23-60/20-32/15-20	Oval, irregular, curved		Authors observation	
Reptile; king cobra; Ophiophagus hannah	46/42/40	Oval		Brachaniec et al. (2022)	
Reptile; Korean ratsnake; Elaphe anomala	21/18/16	Irregular		Brachaniec et al. (2022)	
Reptile; common European viper; Vipera berus	27/12/10	Elongated		Brachaniec et al. (2022)	
Bird; house sparrow; Passer domesticus	14/3/3	Curved		Authors observation	
Bird; city pigeon; Columba livia forma urbana	11/9/7	Oval		Authors observation	
Bird; White-tailed Eagle; Haliaeetus albicilla	250/120/5	Irregular		Brachaniec et al. (2022)	
Mammal; brown hare; Lepus europeaus	13/11/10	Oval		Authors observation	
Mammal; European mole; Talpa europaea	19/5/5	Elonagated, curved		Authors observation	
Mammal; guinea pig; Cavia porcellus	14/6/4	Elongated		Authors observation	
Mammal; clawless; Rollulus rouloul	34/10/10	Curved, irregular		Authors observation	
Mammal; swinhoe’s striped squirrel; Tamiops swinhoei	16/5/5	Elongated, curved		Authors observation	
Mammal; Seba’s short-tailed bat; Carollia perspicillata	12/6/5	Elongated, curved		Authors observation	
Mammal; Eurasian beaver; Castor fiber	14/12/12	Oval		Brachaniec et al. (2022)	
Mammal; African lion; Panthera leo	100/72/68	Irregular		Brachaniec et al. (2022)	
Mammal; Cheetah (Acinonyx)	114/13/10	Elongated, curved		Brachaniec et al. (2022)	

Discussion

Oligocene marine coprolites

Majority of the currently documented coprolites come from the Oligocene sediments of the Menilite-Krosno Series of the Outer Carpathians in southern Poland (for details see Table S1). Bajdek & Bieńkowska-Wasiluk (2020) argued that the high abundance of mesobathypelagic fish remains documented in these sediments may point to a well-oxygenated deep-marine environment (likely exceeding 500 m in some places). Kotlarczyk et al. (2006) concluded that the basin depth in this area could have been even greater, locally exceeding 2,000 m. The coprolites from these deep marine facies were classified into five morphotypes. The first type, characterized by a sinusoidal shape, was previously recorded from Oligocene strata in southern Poland (Bajdek & Bieńkowska-Wasiluk, 2020). These authors concluded that these coprolites were produced by fish predators, mainly representatives of Palimphyes, Oligophus, and an indeterminate gadiform. However, current experimental studies suggest that similar faecal morphologies could also be associated with invertebrates, such as crabs, whose body-fossils remains are relatively common in the Menilite-Krosno Series (Jerzmańska, 1967; Bieńkowska-Wasiluk, 2010; Fig. S7). Although Bajdek & Bieńkowska-Wasiluk (2020) considered crabs as potential producers, they ultimately ruled them out, reasoning that the crabs known from these strata were too small to produce long, sinusoidal coprolites. Noteworthy, the lengths of faecal strings may approach the body lengths of their producers. Furthermore, when estimating producer size, the total faecal mass or the diameter of the coprolite may serve as more reliable indicators of the producer’s body size or anus size, respectively, than the length of faecal strings (see Donovan, 1994). Our experimental studies demonstrate that crabs are capable of producing long faecal strings with sinusoidal morphologies comparable to those observed in the studied fossil coprolites (cf., Figs. 2A–2E and 6K).

We suggest that the three successive morphotypes, i.e., straight, curved with macroscopically visible vertebrate remains, and S-shaped, were produced by fish (see Figs. S2–S5). Morphologically similar non-spiral coprolites (e.g., Figs. 2F–2J) are known from the Eocene deposits of the Green River Formation (Edwards, 1976), the Coldwater Beds (Wilson, 1987), and Messel (Richter & Wedmann, 2005). Rope-like (non-spiral) faecal masses are commonly produced by extant teleost fishes (see Fig. 6N), representatives of which inhabited the Oligocene marine environments in southern Poland. Bajdek & Bieńkowska-Wasiluk (2020) illustrated this type of non-spiral and rope-like coprolites from Oligocene sediments of southern Poland and claimed that they were produced by teleost fish, although they did not specify which taxa would be responsible for their formation (comp. e.g., fig. 2i in Bajdek & Bieńkowska-Wasiluk (2020)). Furthermore, our experimental studies indicate that barracudas may produce more or less regular faecal strings, sometimes terminating in a slightly tapering end (cf., Fig. 6I). It is noteworthy that Kotlarczyk et al. (2006) also reported the presence of barracudas in the Polish Carpathians.

Identifying the producer of the oval coprolite (Fig. 2U) is challenging. None of the marine taxa known from the Menilite-Krosno Series sediments could be easily linked to this morphology based on current experimental results. However, the morphology and size of the coprolite resemble, to some extent, the excrements of some birds, particularly pigeons. Noteworthy, the remains of these birds have been reported from Carpathian sediments (Bocheński, Tomek & Świdnicka, 2010). However, before this interpretation can be further substantiated, a thorough taphonomic analysis of the preservation pathway of bird faeces in marine deposits is required. Bocheński, Tomek & Świdnicka (2010) also reported fossils of humming birds and some passerines from the same strata. However, the shape and size of the faeces of these taxa differ from those of the studied coprolites (Bocheński & Bocheński, 2008; Bocheński et al., 2011; see Fig. 6F).

Miocene marine coprolites

Four coprolites were recorded in the marine Miocene sediments (for details see Table S4). Two of them (GIUS 10–3796/M/13, 32; Figs. 3E, 3F) come from shallow marine deposits displaying high variation of lithologies, facies, and thicknesses (Roztocze area and southern edge of the Holy Cross Mountains). There have been no omnivorous or predatory vertebrates documented in the Żelebsko quarry (Roztocze area) that could have been responsible for the production of the documented coprolites (Wysocka, Jasionowski & Peryt, 2007) and literature cited therein). The dominant species at the site are small gastropods, bivalves, and foraminifers. However, fossil fish teeth are common in a nearby Gołuchów quarry exposing the sediments of the same age (southern edge of the Holy Cross Mountains). These fossils co-occur at the site with fossils of invertebrates, including foraminifers, molluscs, bryozoans, serpulids, echinoderms (asteroids, echinoids and stalked crinoids (Salamon et al., 2024). Most of the fish teeth at the site represent teleost fish (above 70% collected specimens; Salamon et al., 2024). They belonged to the family Sparidae. There have been also shark teeth, but those were less numerous, and belonged mainly to the Odontaspididae family, including Carcharias acutissima and Araloselachus cf. vorax. Salamon et al. (2024) also documented shark teeth (68% of all specimens), belonging to at least four families, in the nearby locality of Zygmuntów near Książ Wielki (see fig. 2 in Salamon et al., 2024). Fossil teeth assigned to Otodus megalodon, Cosmopolitodus hastalis, Isurus, and Galeocerdo were found there as well; myliobatoid teeth were also occasionally noted (Aetobatus). According to Salamon et al. (2024) teleost fish teeth and tooth plates constitute 24% of the collected teeth specimens, and are represented only by Sparidae. A logical step in the challenging task of producer identification would be to seek potential candidates among predatory taxa represented by fossil teeth. The identification, however, is further complicated by the absence of recognizable faunal remains within the coprolite matrix. The list of potential producer candidates can be even longer as other predatory vertebrates (toothed and toothless cetaceans, porpoises) have been recognized in the northern (Polish) part of Miocene Paratethys (Czyżewska & Radwański, 1991 and literature cited therein). These mammals cannot be excluded as the potential producers of coprolites from Żelebsko and Gołuchów. Bałuk (1977) documented numerous remains of cephalopods within the Korytnica Clays of the southern margin of the Holy Cross Mountains. However, the morphology of fossil and extent faeces assignable to these invertebrates (comp., Knaust & Hoffmann, 2020), and literature cited therein) differ from the coprolites from Żelebsko and Gołuchów.

Two coprolites (GIUS 10–3796/M/33, 34; Fig. 3D) have been collected from the Menilite-Krosno Series (The Outer Carpathians, Poland) –strata representing marine environment, probably exceeding 500 m depth (Bajdek & Bieńkowska-Wasiluk, 2020). These are relatively long and complex faecal masses, each consisting of string with frequent constrictions. These features make them similar to the faeces of extent sea cucumbers and cephalopods (see fig. 6, 7 in Knaust & Hoffmann, 2020; Figs. 6L, 6M). However, holothurians have not been described so far from the Menilite-Krosno Series, and only a single cephalopod (Aturia sp.) specimen has been described from the strata (Świdnicka, 2007). Therefore, identification of potential producers must remain speculative as body fossil record is missing or not sufficient.

Miocene continental coprolites

There are excrement-like masses (pellets) that are frequently recorded from various clayey sediments (for review see Brachaniec et al., 2022). However, some researchers rule out zoological origin of those pellets, despite their superficial similarity to faecal masses. The main characteristics cited against the biological origins of those, are: their ferruginous composition, variation in size, lack of internal inclusions, and scarcity of associated (embedded) vertebrate remains (e.g., Roberts, 1958; Dake, 1960; Danner, 1994; Danner, 1997; Spencer & Tuttle, 1980; Love & Boyd, 1991; Spencer, 1993; Spencer, 1997; Hardie, 1994; Mustoe, 2001).

Several hypotheses have been proposed to explain the origin of these problematic masses, including: co-seismic lique faction, sediment intrusion into hollow logs or between plant stems, expulsion of sediment under gravitational pressure, and siderite extrusion driven by methanogenesis (Spencer & Tuttle, 1980; Love & Boyd, 1991; Spencer, 1993; Peterson & Madin, 1997; Mustoe, 2001). However, there have been also a few authors who interpreted these masses as biological in nature, either as fossil faeces (coprolites), cololites, or evisceralites (Amstutz, 1958; Broughton, Simpson & Whitaker, 1977; Broughton, Simpson & Whitaker, 1978; Seilacher et al., 2001; Broughton, 2017; Brachaniec et al., 2022). Recently, Brachaniec et al. (2022) presented a detailed study of excrement-shaped ferruginous masses from the Miocene strata of Poland (Turów, south-west Poland). The authors described two morphotypes: the first includes small, sausage-shaped specimens, while the second comprises larger, more rounded to oval, massive specimens with a rough surface, sometimes exhibiting a prominent pointed end covered by a striated pattern, interpreted as a morphology resulting from anal contraction during the cutting off of the expelled portion of the faecal mass. The latter authors combined their palaeontological and mineralogical analytical results with experimental data and concluded that these structures may represent “true” coprolites, which were likely produced by reptiles (smaller morphotype—by tortoises (Testudinoidea)) and larger one—by snakes (Serpentes)]. This conclusion was supported by the morphological match between the fossil and experimental faecal masses (including fine striations), as well as by the presence of hair-like structures (or coalified inclusions) within the coprolites, which could suggest a diet including mammals. In the current study (see Table S4) we documented thirty (30) ferruginous coprolites (GIUS 10–3796/M/1–12, 14–31). These specimens have been collected from two regions of southern Poland (the Turów area and the Kleszczów Graben area). All these coprolites are represented by one morphotype only (II morphotype sensu Brachaniec et al., 2022; i.e., more rounded to oval, elongate, massive specimens with rough surface; Figs. 3G–3L, 3N). These coprolites comprise numerous hair-like structures, coalified inclusions, and traces of fine striations visible on the surfaces. These features make them similar to other Miocene coprolites ascribed so far to snakes (fig. 2H–M in Brachaniec et al., 2022). However, other producers cannot be ruled out definitively at this stage. A rich assemblages of continental tetrapod fauna have been documented from slightly older sediments (Eocene and Oligocene) of surrounding areas (north-western Bohemia and south-eastern Germany). Brachaniec et al. (2022) mentioned other fossil representatives, including frogs, salamanders, choristoderans, crocodiles, turtles, lizards, and snakes from these regions (for details see table 1 in Brachaniec et al., 2022; see also Górka et al., 2025). The same authors noted that vertebrate fossil remains are abundant in the Miocene of northern Bohemia (North Bohemian Brown Coal Basin in Czechia), and are represented by osteichthyan fish, amphibians, reptiles, birds, and mammals, among others (for details see table 2 in Brachaniec et al., 2022).

Rodents of Sciuridae family could be responsible for the apatite coprolites with a characteristic and prominent pointed termination, that likely formed due to contraction of anus closing to cut off the faecal mass (GIUS 10–3796/M/6, 6(1), 6(2), 6(3), 6(4), 6(5); Fig. 3M). Such coprolites have been found in the sediments of the Kleszczów Graben area (Garapich, 2002; Kowalski & Rzebik-Kowalska, 2002). Chame (2003) studied excrements of extant mammals and illustrated small (max. length 1.5 cm) faeces, with a narrowing termination (see table 1 in Chame, 2003). This type of faeces was produced by Sciuridae (Chame, 2003). Alternatively, it is also possible that representatives of Chiropteridae produced this type of coprolites from the Kleszczów Graben—indeed their fossil remains in the strata have been documented by Garapich (2002; see also Fig. S7).

The current actualistic studies show that the bat (Carollia perspicillata) may produce elongated faeces with a characteristic prominent pointed end formed during anus closing (Fig. 6E). The bat faeces resemble some of the studied fossil specimens (cf., Fig. 3M). Based on the combination of morphology and size, we exclude the possibility that this type of coprolite was produced by representatives of Talpidae, Castoridae, Caviidae, or lizards, despite the presence of their fossils in the sedimentary strata of the Kleszczów Graben (Garapich, 2002 and literature cited therein; comp. Fig. 6, Fig. S7, and data presented in Brachaniec et al., 2022).

Other groups of organisms recorded from this area are malacofauna (Stworzewicz, 1999), fish (Kovalchuk et al., 2019), and crustaceans (Dumont et al., 2020). During fieldwork, we documented also other co-occurring fossils represented by bone elements, vertebrae, teeth, and otoliths of freshwater fish belonging to Gobioidae, Umbridae, Cyprinidae, Pleuronectoidae, Apogonidae and “Anguilloides” sp. (an extinct relative of an eel; Fig. S7). However, the robust morphology (including the pointed termination) and size make representatives of these groups rather unlikely candidates for the producers of the studied ferruginous coprolites.

Supplemental Information

Supplemental Information 1 Examples of crab fossils representing Liocarcinus oligocenicus from the Oligocene marine strata of the Menilite-Krosno Series (The Outer Carpathians, Poland)

(A) Kr.J-7. (B) Kr.H-1. (C) Kr.JR-2. (D) Kr.J-3. (E) Kr.J-11. (F) Kr.J-16. (G) Kr.J-12. (H) Kr.J-6. (I) Kr.J-3. Scale bar equals one cm.

Supplemental Information 2 Examples of fossil fish from the Oligocene marine strata of of t he Menilite-Krosno Series (T he Outer Carpathians, Poland)

(A) Specimen representing u nidentified taxa, Ma 31. (B) Clupea sp., ROJ-215. (C) Specimen representing u nidentified taxa, ROJ-212. (D) Specimen representing u nidentified taxa, ROL-305. (E) Specimen representing u nidentified taxa, ROJ-307. (F) Eomyctophum sp., Ma-52. (G) Holosteus sp., ROJR-170. (H) Unidentified taxa of Scombridae family, ROL-47. (I) Centriscus sp., ROJ-514. (J) Argyropelecus sp., ROL-221. (K) Hipposyngnathus sp., ROJ-211. (L) Specimen representing u nidentified taxa, ROL-328. Scale bar equals 1 cm.

Supplemental Information 3 Examples of fossil fish from the Oligocene marine strata of the Menilite-Krosno Series (T he Outer Carpathians, Poland)

(A) Holosteus sp., ROJ-17. (B) Holosteus sp., ROJ-22. (C) Holosteus sp., ROJ-45. (D) Oligoserranoides sp., ROR-153. (E) Oligoserranoides sp., ROJ-47. (F) Oligoserranoides sp., RORR-7. Scale bar equals 1 cm.

Supplemental Information 4 Examples of fossil fish collected in Oligocene marine strata of the Menilite-Krosno Series (T he Outer Carpathians, Poland)

(A) Scopeloides sp. GIUS10–3796/O/F1. (B) Jaw of Lepidopus sp. (C, D) probably Scopeloides sp. GIUS10–3796/O/F3, 4. Scale bar equals 1 cm.

Supplemental Information 5 Examples of fossil fish from the Oligocene marine strata of the Menilite-Krosno Series (T he Outer Carpathians, Poland)

(A) Lepidopus sp., ROU-400. (B) Lepidopus sp., ROU-405. (C) Lepidopus sp., Ma-5. (D) Lepidopus sp., ROU-40. (E) Lepidopus sp., ROU-42. (F) Lepidopus sp., ROL-55. (G) Isurus sp., ROM-ZR-100.(H) Isurus sp., ROM-ZR-103. (I) Isurus sp., ROM-ZR-107. (J) Isurus sp., ROM-ZR-112. (K) Isurus sp., ROJ-ZR-123. Scale bar equals 1 cm.

Supplemental Information 6 Examples of feathers representing unidentified taxa from the Oligocene marine strata of the Menilite-Krosno Series (T he Outer Carpathians, Poland)

(A) MSMD.Av. Jam-11. (B) MSMD.Av. Jam-14. (C) MSMD.Av. S.Bir-3. (D) MSMD.Av. Jam-1. (E) MSMD.Av. J.Ros-9. (F) MSMD.Av. Jam-15. Scale bar equals 1 cm.

Supplemental Information 7 Some examples of vertebrate remains documented in Miocenian deposits of the Kleszczów Graben, central Poland. Acronyme number: GIUS 10–3796V

(A) Jaw of a Lacertidae lizard. (B) Otolith of Klingobius andjelkocae. (C, D) Vertebrae of indeterminated rodents. (E–H) Bones of indeterminate vertebrates. (I) Jaw of a rodent. (J) Tooth of Chiroptera. (K) Incisor of Castocrinae. (L–N) Talpidaeteeth. (O) Tooth of an unidentified predator. Scale bar equals 1 mm.

Supplemental Information 8 Oligocene coprolite list

Supplemental Information 9 Miocene coprolite list

Supplemental Information 10 Oligocene localities with coprolites and their morphologies

Supplemental Information 11 Miocene localities with coprolites and their morphologies

Supplemental Information 12 A movie showing the 3D-reconstructed internal content ofa selected Oligocene (Rupelian) coprolite of the Kąkolówka locality, southern Poland (specimen no. GIUS 10–3796/O/9)

Journal reviewers and editor are acknowledged for improving the quality of this study. Special thanks are due to all employees and authorities of Municipal Zoological Garden in Łódź, Poland, thanks to whom it was possible to carry out actualistic research. Authors thanks to Tomasz Janiszewski for photos of bird faeces, zoological shop “Rybka” for fish faeces, Robert Szybiak for lending some of the specimens for research, and Wojciech Krawczyński for photo documentation. Maciej Pindakiewcz and Adiel A. Klompmaker are acknowledged for otoliths and crabs assignment.

Additional Information and Declarations

Competing Interests

Author Contributions

Data Availability

The authors declare there are no competing interests. Adam Danielak and Magdalena Janiszewska are employed by Municipal Zoological Garden in Lódz (Polish limited liability company).

Tomasz Brachaniec conceived and designed the experiments, performed the experiments, analyzed the data, prepared figures and/or tables, authored or reviewed drafts of the article, and approved the final draft.

Dorota Środek performed the experiments, analyzed the data, prepared figures and/or tables, authored or reviewed drafts of the article, and approved the final draft.

Mateusz Salamon performed the experiments, prepared figures and/or tables, authored or reviewed drafts of the article, and approved the final draft.

Michał Bugajski performed the experiments, analyzed the data, prepared figures and/or tables, authored or reviewed drafts of the article, and approved the final draft.

Piotr Duda performed the experiments, prepared figures and/or tables, authored or reviewed drafts of the article, and approved the final draft.

Adam Danielak performed the experiments, authored or reviewed drafts of the article, and approved the final draft.

Magdalena Janiszewska performed the experiments, authored or reviewed drafts of the article, and approved the final draft.

Grzegorz Sadlok performed the experiments, analyzed the data, authored or reviewed drafts of the article, and approved the final draft.

Wojciech Kuśnierczyk performed the experiments, analyzed the data, prepared figures and/or tables, authored or reviewed drafts of the article, and approved the final draft.

The following information was supplied regarding data availability:

The data is available at Zenodo: Tomasz Brachaniec, Dorota Środek, Mateusz Salamon, Michał Bugajski, Piotr Duda, Adam Danielak, Magdalena Janiszewska, Grzegorz Sadlok, & Wojciech Kuśnierczyk. (2025–2026). Image stacks and software for viewing them. In From Trace to Trace Maker: Oligocene–Miocene Coprolites of Southern Poland and Their Potential Producers. Zenodo. https://doi.org/10.5281/zenodo.16742330.

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
