# Peer review of "From trace to trace maker: Oligocene–Miocene coprolites of southern Poland and their potential producers"

_PeerJ, doi:10.7717/peerj.20242_

## Round 0.1 · original submission · Major Revisions

You provide important new data on coprolite from the Miocene and Oligocene of Poland as well as provide crucial documentation of modern faeces. I therefore would like to see this published, but the manuscript needs crucial revisions before publication. The main point are:

Structure: I agree with reviewers 2 and 3 that the structure needs substantial modification. In the introduction, it shall more clearly be stated the scientific questions/hypotheses you want to answer (compare reviewer 2), the results should focus on describing the different morphotypes (or subtypes) of coprolites (ideally designated as letters and/or numbers: compare Rozada et al. 2024) and modern faeces without interpreting their assignment (compare reviewer 3 and 1). In the discussion particular morphotypes should be assigned to taxa which potentially produced them based on similarity in morphology/inclusions (compare reviewer 3) and context (Francischini et al. 2018) as well as subquently implications for interpreting the paleoenvironments should be discussed (compare reviewer 2).

Table: Please add a table with different morphotypes, their age/locality, the putative assignment to potential producer and the main argument(s) to do so (e.g., morphology, composition, inclusions, actualistic comparisons, etc.) would make the work and your interpretations easier to follow.

Modern comparisons: The description and documentation of the modern faeces could be expanded (compare reviewer 3). Modern faeces is rarely figured so any new documentation is valuable particularly when accompagnied by a description of their morphology, assignment and context (e.g., Dentzien-Dias et al. 2018). Also homogenizing the depictions of modern faeces would also be helpful.

Ecology: Although I feel that assigning coprolite to morphotypes and assigning them to potential producer is valuable in its own right particularly in comparison with modern producers, I agree with reviewer 2 that providing implications for the paleoecology of the environments would also be helpful in this context.

Used techniques: please explicitly on which approach/techniques you were able to assign/interpret texture, composition, minerals and origin of the phosphate and also provide the associated data (compare reviewer 3). The use of colour alone is insufficient to resolve composition. Please also take into account how taphonomy may have altered the morphology and composition (compare reviewer 2 and 3)

Tomographic data: A video is considered insufficient to share raw data for tomographic data. It is necessary to provide at least the raw data (image stack) and it would also be customary to provide the reconstructed 3D model in a widely accessible format for the sake of reproducibility (compare Davies et al. 2017).

Formatting and typographically issues: please make sure to check formatting and language of the manuscript (compare reviewer 3). I advise to let a colleague fluent in English to proofread the revised manuscript (e.g., ambient environment may be better than natural environment given they were derived from animals in captivity, etc.). There are various issues with lack of spaces between words to typographical issues (otoliths not otholits, etc.). Please see the annotated pdf for more examples.

Figures: It may be a matter of preference but there is no need for double scale bars (Figs. 2, 3 and 6). The contrast (e.g, Fig. 6E) and/or resolution (e.g., Fig. 6G) of multiple picture needs be improved. My suggestions would be to cut the original scale bar (“ruler”) as much as possible (e.g., sometimes the scalebar is larger than the actual faeces) keeping the digitally added homogenous scalebar. Also, please consider filling the freed space with reconstructions of photographed faeces (in same cases, their shape is hard to distinguish from the background). I am ok with documenting associated fauna providing extra context particularly when not available in literature (Figures 7–13). However, their relevance should be more clearly justified and described in text. There presence could potentially be partially reduced or moved to supplementary material for conciseness (compare reviewer 3), but I have no strong opinion on this aspect. I understand the convenience of having all in one place rather than scattered in the literature.

Please make sure to address these as well as all other points raised by the reviewers and myself including those in annotated pdfs.

I look forward to receiving the revised manuscript.

Suggested references

Davies, T. G., Rahman, I. A., Lautenschlager, S., Cunningham, J. A., Asher, R. J., Barrett, P. M., ... & Donoghue, P. C. (2017). Open data and digital morphology. Proceedings of the Royal Society B: Biological Sciences, 284(1852), 20170194.

Dentzien-Dias, P., Carrillo-Briceño, J. D., Francischini, H., & Sánchez, R. (2018). Paleoecological and taphonomical aspects of the Late Miocene vertebrate coprolites (Urumaco Formation) of Venezuela. Palaeogeography, Palaeoclimatology, Palaeoecology, 490, 590-603.

Francischini, H., Dentzien-Dias, P., & Schultz, C. L. (2018). A fresh look at ancient dungs: Brazilian Triassic coprolites revisited. Lethaia, 51(3), 389-405.

Rozada, L., Allain, R., Qvarnström, M., Rey, K., Vullo, R., Goedert, J., ... & Robin, N. (2024). A rich coprolite assemblage from Angeac-Charente (France): a glimpse into trophic interactions within an Early Cretaceous freshwater swamp. Cretaceous Research, 162, 105939.

·

Basic reporting

This is an interesting paper reviewing some important Miocene and Oligocene coprolites from Poland. I have made some suggestions on the manuscript including: (1) describe in detail the main morphotypes when they are first mentioned; (2) discriminate the morphotypes in the figure descriptions of Figs. 2-3; and (3) I think you meant to cite Hunt and Lucas 2021 not 2012 as the latter does not describe the fossil record of coprolites.

Experimental design

No comment

Validity of the findings

No comment

Additional comments

I hope the authors will publish a long description of the recent faeces samples.

·

Basic reporting

This manuscript contributes useful descriptive data to the field of coprology, particularly through a wide-ranging survey of post-Mesozoic coprolites across diverse depositional environments in Poland. Drawing upon the authors’ prior works, I would like to raise several concerns regarding the broader scientific implications of the findings.

Overall, the manuscript is clearly written and understandable, though some sections could benefit from minor linguistic polishing. My primary concern, however, lies not in language but in the scientific significance of the work. Simply reporting coprolite morphotypes and proposing potential producers has limited value unless the interpretations are supported by strong evidence and a clear ecological framework. Greater emphasis should be placed on how the coprolite assemblages inform our understanding of the paleoecology of these environments.

Experimental design

The most potentially impactful element of this study is the inclusion of a large comparative dataset of both fossil and extant excrement. However, the paper does not clearly distinguish how many of the extant feces examined are new additions relative to previous studies, especially those reported by Brachaniec et al. (2022).

The current manuscript appears to function more as a compilation of coprolite specimens from various localities, which, while extensive, feels loosely connected from a scientific hypothesis or question standpoint. A more cohesive methodological framework—ideally quantitative—would enhance the value of the comparative actualistic approach.

Additionally, although the study of extant feces is commendable, the correlation between modern and fossil material must be approached with caution. It remains unclear how convincingly the morphological similarities between modern feces and ancient coprolites can support taxonomic attribution without taphonomic consideration.

Regarding figures: Figures 7–13 seem excessive and contain a wide range of body fossils that are not always directly tied to the main focus of the paper. Their relevance should be more clearly justified or reduced for conciseness.

Validity of the findings

My main concern pertains to the validity of the conclusions drawn from the presented data. Sideritic and ferruginous masses, especially those interpreted as snake coprolites, remain controversial in the literature. Their biogenicity should be more rigorously defended, with clearer exclusion of alternative (abiotic) origins.

The hair-like inclusions described are interesting but require stronger interpretation—ideally supported by histological or chemical analysis—to confirm their biological origin.

The CT scan data should be presented in a more transparent and interpretable form. Reconstructions must be clearly visible and convincing if they are to substantiate claims regarding internal inclusions or producer attribution.

Additional comments

Many of my concerns align with those expressed by:
• Mustoe, G. (2022). Peer Review #1 of "Comparative actualistic study hints at origins of alleged Miocene coprolites of Poland (v0.1)". PeerJ. https://doi.org/10.7287/peerj.13652v0.1/reviews/1
• Dentzien-Dias, P. (2022). Peer Review #2 of "Comparative actualistic study hints at origins of alleged
Miocene coprolites of Poland (v0.1)". PeerJ. https://doi.org/10.7287/peerj.13652v0.1/reviews/2

In summary, while this study presents a broad and diverse dataset, its current structure, interpretations, and scope require refinement to elevate it from a descriptive catalogue to a more hypothesis-driven, ecologically meaningful contribution.

Reviewer 3 ·

Basic reporting

This manuscript makes a nice contribution to the uncommon report of fossil feces assemblages from the deep time. It also provides totally usefull new imaging or rarely reported dejections of modern animals.
It constitutes a truely itneresting contribution to Peer J scientific articles joining fossil and modern ecological novel data.

My overall aprreciation goes to rework the description and producer assignation process in a more robust way, which I detailed further and in the pdf. As of now, it gives an impression of "too quickly assigned" which shall be avoid given that will be further likely much cited (eg: pigeon, barracuda or sea cuncumber feces). The english is good, just locally too long or with small mistakes (see pdf). The care brought to the results and caption parts is to be improved in my opinion. Also modern occurences are too poorly figured for such journal - please reshape.

There are many phrasing clumsiness and errors in interpreting coprolite material at high scale (texture, composition, minerals, origin of the phosphate). I think they shall explain us how they identified fluorapatite and provide the corresponding data (eds, images, diffractometry? etc)

Experimental design

The authors often refer to the "experimental studies" refering to images of faeces collected from animals kept in vivo. However there approach clearly lack details on

- how they selected some taxa to collect their modern feces and if they didn't check at others that were reported as fossils from the same locality

- if they are aware that terminal ends of feces can much change depending on matrix hardness (and then diet change).

I really encourage them to better describe in results the shapes of the modern feces that they here use to support their assignation reasoning.

On top of that, they shall better describe the different morphotypes that they did recognize cause descriptions as strict shape names as such can totally overlay one another.

Validity of the findings

It feels like authors reached the fact that they could not really conclude on most coprolite assignation but developed too long discussion to arrive to this conclusion. I would rather recommend to describe in results:
- fossil morphotypes
- modern results

And only in disucssion: take fossil cases one by one as currently and develop similarities between modern and fossils with much more details; and above all, to focus first on EXCLUDING what certainly can not have produced these fossils.

A very explicit case, is the consideration of taxa that produce strictly spiral coprolites (elasmobranchs, holosteans, polypters) for non-spiral coprolites - it gives an impression of way too little attention to the fossil material description prior to interpretation based on what was in the environnement and what new pictures of modern feces show. But these are necessarily highly non-exhaustive !

Therefore, the findings are interesting but shall in my opinion be more temperate at the only light of what can only be said taking into account the outside and inside.

Additional comments

I provided most the help I could to improve the content in a more structured and more straightforward way in the attached .pdf. I really think that the authors shall work further on it, but I very much encourage the target and interest of the manuscript. All the best with the improvement phase.

Annotated reviews are not available for download in order to protect the identity of reviewers who chose to remain anonymous.

---

## Round 0.2 · Minor Revisions

Thank you for addressing our suggestions in the revised manuscript. I apologize for delay in making my decision, but given the manuscript´s length, I needed some more time to check it one more time fully before making it. The neoichnological approach is highly commendable and the revision makes the manuscript even easier to follow and of broader relevance. Making available the raw data of the tomographic analyses is also good standard practice making it easier to reproduce your results. I would love to see this published, and I feel we are close to acceptance, but there are still some points I would like you to address before. The main points being:

Morphotype M4 (Line 335 – 340): the current description of this morphotype makes it seem like a loose grouping of morphologies which could not be assigned to other morphotypes. I feel this ok if this is the case, but it should likely be phrased even more clearly if this is the case. Otherwise, I would be crucial to list why all these seemingly different morphologies were assigned to the same morphotype.

Assignment to modern taxa of producers: As I stated previously and agreeing with reviewer 2, I greatly appreciate your neoichnological approach and comparison with modern faeces. In some case, it would like to see an even more cautious approach (e.g., please entertain expanding “produced by barracudas” to “produced by barracudas or related fishes” or “produced by barracudas or other fishes with similar behaviour/anatomy”). Could you provide some additional details on producer of non-spiral coprolites you mentioned on line 511-512 (e.g., were they also interpreted to be produced by fishes). I would drop “coprolite” before “morphotypes” on line 590 as you only interpret these masses as coprolites after this line. Please refer to associated fauna figured in the supplementary figures (e.g., line 637 and other places where relevant).

In addition, there are still some formatting and language issues which need to be resolved:

Line 139: “might be an” instead of “might bean”

Line 378: do really mean “cementation” or rather “mineralization”

Line 463: “by barracuda” instead of “bybarracuda”

Line 471: “feces” instead of “faeces”; please make sure to consistently use feces throughout the text

Line 590: drop “coprolite” in front of morphotypes

Line 684-685: there seem to various typographical issues in this sentence: please use “acknowledged” instead “ackowledged”; “otoliths” instead of “otholits”; “assignment” instead of “assignation”

Please make sure to address these as well as other points raised by me or the reviewers (if any). Please ask a colleague fluent in English to read the manuscript and check English one more time as I am also not native speaker.

I look forward to receiving the revised manuscript.

·

Basic reporting

I think the manuscript revisions adequately address the majority of the editorial and reviewer comments.

Experimental design

I think the manuscript revisions adequately address the majority of the editorial and reviewer comments.

Validity of the findings

I think the manuscript revisions adequately address the majority of the editorial and reviewer comments.

Additional comments

I think the manuscript revisions adequately address the majority of the editorial and reviewer comments.

·

Basic reporting

The research article describes coprolites from deep-marine Oligocene sediments, shallow- and deep-marine Miocene deposits, as well as Miocene continental environments in southern and central Poland.

The authors have made a commendable effort to clarify most of the questions raised during the first round of review. The manuscript is now more organized, scientifically refined, and easier to comprehend.

The figures have been improved and streamlined, with a stronger emphasis on those most relevant to the main objectives of the study.

There remain some minor issues with language, punctuation, and title formatting, which can be addressed through careful revision with the authors’ attention.

Experimental design

The authors have provided relevant responses to the issues raised regarding the experimental design, and I find their arguments convincing.

They have improved the explanation of their methodology, making the text clearer and easier to follow.

The application of neoichnological approaches to coprolite studies is highly commendable, and this has been effectively implemented in the present work.

For future research, the authors may also consider incorporating ichnotaxonomic analyses to further strengthen their interpretations.

Validity of the findings

The authors have provided relevant responses to the issues raised regarding the validity of the findings, and I find their arguments convincing.

Figure 6 and Table 1 present some of the most important and informative data in the manuscript.

For future studies, the authors may consider reconstructing the paleoecological implications of the localities based on coprolite evidence, which would add greater depth to the interpretations.

Additional comments

-no comment-

---

## Round 0.3 · accepted · Accept

Thank you for addressing these final suggestions which make the manuscript even easier to follow, understand and reproduce. I agree that all the reviewer suggestions were properly addressed. I am therefore happy to recommend this manuscript for publication and look forward to seeing it published. I only found one typographical issue (line 453: “Carpathiansno” should be replaced by “Carpathians”) which should be corrected during the proofing phase. In the cases I checked the morphotypes seems to have been consistently referenced in text and figure captions, but please check it on more time during the proofing phase.